# ORDERED GNN: ORDERING MESSAGE PASSING TO DEAL WITH HETEROPHILY AND OVER-SMOOTHING

**Yunchong Song**[1], **Chenghu Zhou**[2], **Xinbing Wang**[1], **Zhouhan Lin**[1*]
[1]Shanghai Jiao Tong University, [2]Chinese Academy of Sciences
{ycsong, xwang8}@sjtu.edu.cn, lin.zhouhan@gmail.com

## ABSTRACT

Most graph neural networks follow the message passing mechanism. However, it faces the over-smoothing problem when multiple times of message passing is applied to a graph, causing indistinguishable node representations and prevents the model to effectively learn dependencies between farther-away nodes. On the other hand, features of neighboring nodes with different labels are likely to be falsely mixed, resulting in the heterophily problem. In this work, we propose to order the messages passing into the node representation, with specific blocks of neurons targeted for message passing within specific hops. This is achieved by aligning the hierarchy of the rooted-tree of a central node with the ordered neurons in its node representation. Experimental results on an extensive set of datasets show that our model can simultaneously achieve the state-of-the-art in both homophily and heterophily settings, without any targeted design. Moreover, its performance maintains pretty well while the model becomes really deep, effectively preventing the over-smoothing problem. Finally, visualizing the gating vectors shows that our model learns to behave differently between homophily and heterophily settings, providing an explainable graph neural model.[1]

## 1 INTRODUCTION

Graph neural network (GNN) (Scarselli et al., 2008; Bruna et al., 2013; Defferrard et al., 2016; Kipf & Welling, 2016; Veličković et al., 2017; Hamilton et al., 2017; Gilmer et al., 2017; Xu et al., 2018a) has become the prominent approach to learn representations for graphs, such as social networks (Li & Goldwasser, 2019), biomedical information networks (Yan et al., 2019), communication networks (Suárez-Varela et al., 2021), n-body systems (Kipf et al., 2018), etc. Most GNNs rely on the message passing mechanism (Gilmer et al., 2017) to implement the interactions between neighbouring nodes. Despite its huge success, message passing GNNs still faces two fundamental but fatal drawbacks: it fails to generalize to heterophily where neighboring nodes share dissimilar features or labels, and a simple multilayer perceptron can outperform many GNNs (Zhu et al., 2020b), this limit GNNs extending to many real-world networks with heterophily; it is also observed the node representations become indistinguishable when stacking multiple layers, and suffers sharply performance drop, resulting in the so-called "over-smoothing" problem (Li et al., 2018), which prevent GNNs to utilize high-order neighborhood information effectively.

To address these two drawbacks, numerous approaches have been proposed. Most of them concentrate on the aggregation stage of message passing. Some design signed messages to distinguish neighbors belong to different classes (Yang et al., 2021; Bo et al., 2021; Luan et al., 2021; Yan et al., 2021), allowing GNNs to capture high-frequency signals; Min et al. (2020) design specific filters to capture band-pass signals; some apply personalized aggregation with reinforcement learning (Lai et al., 2020) or neural architecture search (Wang et al., 2022b); others attempt to aggregate messages not only from the direct neighbors, but also from the embedding space (Pei et al., 2020) or higher-order neighbors (Zhu et al., 2020b). These aggregator designs have achieved good performance, however, they primarily focus on the single-round message passing process and ignore the integration of messages from multiple hops. Another line of works focus on the effective utilization

---

[*]Zhouhan Lin is the corresponding author.

[1]Code is available at https://github.com/LUMIA-Group/OrderedGNN

of multiple hops information, which is mainly accomplished by designing various skip-connections. Klicpera et al. (2018); Chen et al. (2020b) propose initial connection to prevent ego or local information from being "washed out" by stacking multiple GNN layers; inspired by ResNet (He et al., 2016), some works (Li et al., 2019; Chen et al., 2020b; Cong et al., 2021) explores the application of residual connection on GNNs to improve the gradients; others combined the output of intermediate GNN layers with well-designed components, such as concat (Xu et al., 2018b; Zhu et al., 2020b), learnable weights (Zhu et al., 2020b; Abu-El-Haija et al., 2019; Liu et al., 2020), signed weights (Chien et al., 2020), or RNN-like architectures (Xu et al., 2018b; Sun et al., 2019). These works are simple yet effective, however, they can only model the information *within few hops*, but unable to model the information exactly *at some orders*, this lead to a mixing of features at different orders; besides, many of these approaches (Li et al., 2019; Abu-El-Haija et al., 2019; Chen et al., 2020b; Zhu et al., 2020b; Chien et al., 2020; Cong et al., 2021) are unable to make personalized decisions for each node. These deficiencies result in suboptimal performance. In addition to caring about the model side, other approaches focus on how to modify the graph structure. These methods are called "graph rewiring", including randomly removing edges (Rong et al., 2019) or nodes (Feng et al., 2020), or computing a new graph with heuristic algorithms (Suresh et al., 2021; Zeng et al., 2021). In general, these algorithms are not learnable and thus only applicable to certain graphs.

Unlike the previous works, we address both problems by designing the combine stage of message passing and emphasize the importance of it. *The key idea is to integrate an inductive bias from rooted-tree hierarchy, let GNN encode the neighborhood information exactly at some orders and avoid feature mixing within hops.* The combine stage has been rarely focused before, most works simply implement it as a self-loop. This would result in an unreasonable mixing of node features (Zhu et al., 2020b). To avoid this "mixing", Hamilton et al. (2017); Xu et al. (2018b); Zhu et al. (2020b) concat the node representation and the aggregated message, which has been identified as an effective design to deal with heterophily (Zhu et al., 2020b). However, keeping the embedding dimension constant across layers, the local information will be squeezed at an exponential rate. The most related work to ours is Gated GNN (Li et al., 2015), it applys a GRU in the combine stage and strengthen the expressivity, but fails to prevent feature mixing, limiting the performance.

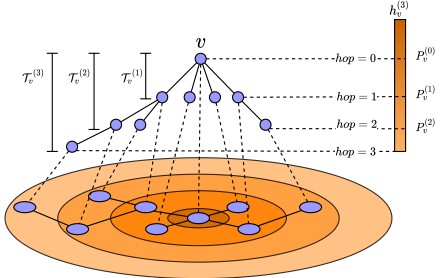

Figure 1: Aligning the hierarchy of a rooted-tree $\mathcal{T}_v^{(k)}$ underlying the graph with the node embedding of the root node $v$. Neighboring nodes within $k$ hops of edges to $v$ naturally form a depth $k$ subtree. Messages passed to $v$ from nodes within this subtree are restricted to the first $P_v^{(k)}$ neurons in the node embedding of $v$.

In this paper, we present the message passing mechanism in an ordered form. That is, the neurons in the node embedding of a certain node is aligned with the hierarchies in the rooted-tree of the node. Here, by rooted-tree of a node we refer to the tree with the node itself as the root and its neighbors as its children. Recursively, for each child, its children are again the neighboring nodes of the child. (c.f. Figure 1). We achieve the alignment by proposing a novel ordered gating mechanism, which controls the assignment of neurons to encode subtrees with different depth. Experimental results on an extensive set of datasets show that our model could alleviate the heterophily and over-smoothing problem at the same time. Our model provides the following practical benefits:

- We design the combine stage guided by the rooted-tree hierarchy, a very general topological inductive bias with least assumption about the neighbors' distribution, this allows a flexible integration of information at different orders, and leading superior performance in both heterophily and homophily.

- The ordered gating mechanism prevent the mixing of node features within hops, enable us to model information at different orders. This open a door to extract similar neighborhood patterns for each node under heterophily; this also make it easy to preserve ego and local information, then effectively alleviating over-smoothing.

- Our model aligns neighboring structures with blocks in node embeddings through an explicit gating mechanism, thus the gating mechanism could provide visualizations to reveal the connecting type of the data and offer explainability.

## 2 OUR APPROACH

In this section, we will first present a general formalization of GNNs. Then we show how to align the hierarchy of the rooted-tree to the node embeddings, the intuitions behind it, as well as the two necessary components to implement it. Finally, our model is presented in Section 2.5, with few cases to explain it's practical benefits.

### 2.1 GRAPH NEURAL NETWORKS

Consider an unweighted and self-loop-free graph $\mathcal{G} = (\mathcal{V}, \mathcal{E})$, with $\mathcal{V}$ and $\mathcal{E}$ denoting its nodes and edges, respectively. Let $v \in \mathcal{V}$ being one of the nodes with its label $y_v \in \mathcal{Y}$, and assume that there are $N$ nodes in the graph. The adjacency matrix for graph $\mathcal{G}$ is denoted as $A \in \mathbb{R}^{N \times N}$. Assume that the input features for each node form a matrix $X \in \mathbb{R}^{N \times F}$, where $F$ is the dimension of input features, with $X_v \in \mathbb{R}^N$ being the features of node $v$. Before feeding it into the GNN, an input projection function $f_\theta(\cdot)$ is optionally incorporated to nonlinearly project the features into a space mathing the node embedding dimensions, i.e.,

$$Z_v = f_\theta(X_v) \tag{1}$$

where $D$ is the dimension of node embeddings, and $Z_v \in \mathbb{R}^D$ is feed to the first layer of GNN, i.e., $h_v^{(0)} = Z_v$. In our work, we simply parameterize $f_\theta(\cdot)$ with an MLP.

Generally, a GNN layer with message passing can be divided into two stages: the *aggregate* stage and the *combine* stage. For the node $v$ at the $k$-th layer, it first aggregates information from its neighbors, forming the message vector $m_v^{(k)}$. Then in the second stage $m_v^{(k)}$ is combined with $v$'s ego representation $h_v^{(k-1)}$ from the last layer. Formally,

$$m_v^{(k)} = \text{AGGREGATE}^{(k)} \left( \left\{ h_u^{(k-1)} : u \in \mathcal{N}(v) \right\} \right), h_v^{(k)} = \text{COMBINE}^{(k)} \left( h_v^{(k-1)}, m_v^{(k)} \right) \tag{2}$$

where $\mathcal{N}(v)$ is the set of node $v$'s direct neighbors. The $\text{AGGREGATE}(\cdot)$ operator denotes a permutation-invariant function on all the neighboring nodes of $v$, such as addition, mean pooling or max pooling, optionally followed by a transformation function. The transformation function could be an MLP (Xu et al., 2018a) or transformation matrix (Kipf & Welling, 2016). The $\text{COMBINE}(\cdot)$ operator denotes a combine function that fuses the ego information and the aggrated message.

Our model only modifies the *combine* stage, which means it could be seamlessly extended to many popular GNNs, such as GCN (Kipf & Welling, 2016), GAT (Veličković et al., 2017), GIN (Xu et al., 2018a), etc., by changing the *aggregate* stage correspondingly. In this paper, we simply implement the *aggregate* stage as computing the mean value of neighboring vectors.

### 2.2 ALIGNING ROOTED-TREE WITH NODE EMBEDDING

From the viewpoint of a node, the neighboring nodes within multiple hops around it can be naturally presented as a rooted-tree (Xu et al., 2018a;b; Liu et al., 2020). For node $v$, its $k$-th order rooted-tree $\mathcal{T}_v^{(k)}$ is composed by putting its first order neighbors as its children, and then recursively for each child, putting the neighboring nodes of the child as its children (except the child that is also the ancester of the node), unitl its $k$-th order neighbors (Figure 1).

Since there is only one round of message passing in each GNN layer, stacking multiple layers of GNNs corresponds to doing multiple rounds of message passing. And since for each single round of message passing, only direct neighbors exchange their information, the node representation of $v$ in the $k$-th layer $h_v^{(k)}$ is only depending on the nodes within its $k$-th order rooted-tree $\mathcal{T}_v^{(k)}$. Our goal is to properly encode the information within the $k$-th order rooted-tree into the fixed size vector $h_v^{(k)}$.

Note that there is a nested structure between rooted-trees of $v$, and we are going to utilize this structure as anchors to align the rooted-trees with the node embedding of $v$. It is obvious that the $k-1$-th rooted-tree $\mathcal{T}_v^{(k-1)}$ is a subtree of the $k$-th rooted-tree $\mathcal{T}_v^{(k)}$, with the same root $v$ and the same tree structure up to depth $k-1$. Thus we have (assuming the model has $K$ layers)

$$\mathcal{T}_v^{(0)} \subseteq \mathcal{T}_v^{(1)} \subseteq \cdots \subseteq \mathcal{T}_v^{(k)} \subseteq \cdots \subseteq \mathcal{T}_v^{(K)}. \tag{3}$$

where $\mathcal{T}_v^{(0)}$ corresponds to the neurons used to represent the node's ego information. As $k$ grows, $\mathcal{T}_v^{(k)}$ becomes larger and more complex, requiring more neurons to encode its information. As a result, it is natural to assume that the neurons used to represent message passing within $\mathcal{T}_v^{(k-1)}$ should be a subset of those for $\mathcal{T}_v^{(k)}$. Correspondingly, we are going to implement the same nested structure in the node embedding of $v$. We first order all the neurons in $h_v$ into a sequence, thus we can index them with numbers. For information within $\mathcal{T}_v^{(k-1)}$, we allow it to be encoded within the first $P_v^{(k-1)}$ neurons. As for the next level rooted-tree $\mathcal{T}_v^{(k)}$, more neurons are involved, corresponding to the first $P_v^{(k)}$ neurons. Apparently here we have $P_v^{(k-1)} \leq P_v^{(k)}$.

There is a critical property that comes with this way of ordering the neurons. Consider the neurons between the two split points $P_v^{(k-1)}$ and $P_v^{(k)}$. They reflect the delta between $k-1$ and $k$ rounds of message passing, encoding the neighbor information exactly *at $k$-th order*. Intuitively, it organizes the message passing in an *ordered* form, avoid the mixing of neighbor features at different orders within a hop. We implement the ordered neurons with the gating mechanism (which will be introduced in the following sections in detail), thus it's natural to visualize the gates as a way to probe the message passing of the model.

Now we have the node embedding of $v$ split by a set of spliting points

$$P_v^{(0)} \leq P_v^{(1)} \leq \cdots \leq P_v^{(k)} \leq \cdots \leq P_v^{(K)}. \tag{4}$$

Each of them is predicted by a GNN layer, with $P_v^{(K)} = D$ pointing to the end of the node embedding. If we can learn the positions of these spliting points and ensure their relative magnitudes, then the alignment could be realized. We are going to show how we predict the spliting points through a gating mechanism, and how we lock their relative magnitudes with the differentiable OR operator. Our model naturally emerges by putting everything together in Section 2.5.

## 2.3 THE SPLIT POINTS

Consider the split point $P_v^{(k)}$ that split up the ordered node embedding into two blocks, with the left block containing neurons indexed between $[0, P_v^{(k)} - 1]$ and the right block $[P_v^{(k)}, D]$. We can represent the split by a $D$-dimensional gating vector $g_v^{(k)}$, with its left $P_v^{(k)}$ entries are 1s and the rest are 0s. This gating vector could then be used in the *combine* stage to control the integration between the ego representation as input to the $k$-th layer $h_v^{(k-1)}$ and the aggregated context $m_v^{(k)}$.

$$h_v^{(k)} = g_v^{(k)} \circ h_v^{(k-1)} + \left(1 - g_v^{(k)}\right) \circ m_v^{(k)} \tag{5}$$

Where $\circ$ denotes element-wise multiplication. Intuitively, for the left $P_v^{(k)}$ neurons in $h_v^{(k)}$, since their corresponding gating vectors in $g_v^{(k)}$ are 1s, they suppress the newly aggregated context $m_v^{(k)}$ and only inherit the last layer's output $h_v^{(k-1)}$. And vice versa for the right block of neurons with gating vectors of 0.

Ideally, we would prefer a crispy split, with a clear boundary at $P_v^{(k)}$ and binary gating vectors in $g_v^{(k)}$. However, this will result in discretized operations, rendering the model non-differentiable. In this work, we resort to "soften" the gates by predicting the expectations of them, thus keeping the whole model differentiable. The expectation vector $\hat{g}_v^{(k)}$ is presented as the cumulative sum of the probability of each position in the node embedding being the split point $P_v^{(k)}$.[2] Concretely, $\hat{g}_v^{(k)}$ is parameterized as

$$\hat{g}_v^{(k)} = \text{cumax}_{\leftarrow} \left(f_\xi^{(k)}\left(h_v^{(k-1)}, m_v^{(k)}\right)\right) = \text{cumax}_{\leftarrow}\left(W^{(k)}\left[h_v^{(k-1)}; m_v^{(k)}\right] + b^{(k)}\right) \tag{6}$$

where $\text{cumax}_{\leftarrow}(\cdot) = \text{cumsum}_{\leftarrow}(\text{softmax}(\ldots))$ is the composite operator of cumulative sum and softmax, with the subscript $\leftarrow$ indicating that the direction of cumulation is from right to left. $f_\xi^{(k)}(\cdot)$ denotes a function fusing the two vectors $h_v^{(k-1)}$ and $m_v^{(k)}$. Here we simply put it as a linear projection with bias $b^{(k)}$. The $\left[h_v^{(k-1)}; m_v^{(k)}\right]$ indicates the concatenation of the two vectors $h_v^{(k-1)}$ and $m_v^{(k)}$.

---

[2]We refer the readers to Appendix A.1 for a proof of $\hat{g}_v^{(k)}$ being the expectation of the gating vector $g_v^{(k)}$.

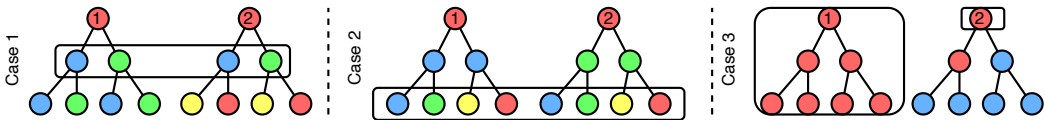

Figure 2: Some rooted-tree cases, each rooted-tree is constructed from root node by message passing, node's color indicate it's label. Case 1 and Case 2 show two cases under heterophily; Case 3 is under homophily, node 1 & 2 are at the center and the border of a community, respectively.

## 2.4 THE DIFFERENTIABLE OR OPERATOR

In the above section we provide a differentiable way to predict the position of the split points, as well as corresponding gates that implement the split operation. However, there's no guarantee that the predicted split points will comply with the restrictions in Equation 4. Not ensuring Equation 4 would break the alignment between the rooted-trees and node embeddings.

To ensure such a relative magnitude, we conduct a bitwise OR operator between the newly computed gating vector $\hat{g}_v^{(k)}$ and its predecessor. We change the notation to $\tilde{g}_v^{(k)}$ to discriminate it from the originally computed gating vector $\hat{g}_v^{(k)}$. Again, the OR operator is not differentiable in nature. We instead implement the softened version of it by

$$\tilde{g}_v^{(k)} = \text{SOFTOR}(\tilde{g}_v^{(k-1)}, \hat{g}_v^{(k)}) = \tilde{g}_v^{(k-1)} + (1 - \tilde{g}_v^{(k-1)}) \circ \hat{g}_v^{(k)} \tag{7}$$

We call it the $\text{SOFTOR}(\cdot)$ operator.

## 2.5 PUTTING IT ALL TOGETHER

With the above discussion, we can get the updating rule for the $k$-th GNN layer by putting everything together, leading to the *Ordered GNN* model:

$$
\begin{aligned}
m_v^{(k)} &= \text{MEAN}\left(\left\{h_u^{(k-1)} : u \in \mathcal{N}(v)\right\}\right) \\
\hat{g}_v^{(k)} &= \text{cumax}_{\leftarrow}\left(f_\xi^{(k)}\left(h_v^{(k-1)}, m_v^{(k)}\right)\right) \\
\tilde{g}_v^{(k)} &= \text{SOFTOR}\left(\tilde{g}_v^{(k-1)}, \hat{g}_v^{(k)}\right) \\
h_v^{(k)} &= \tilde{g}_v^{(k)} \circ h_v^{(k-1)} + \left(1 - \tilde{g}_v^{(k)}\right) \circ m_v^{(k)}
\end{aligned}
\tag{8}
$$

The first equation corresponds to the *aggregate* stage, while the latter three correspond to the *combine* stage. In practice, we can decrease the parameters in the gating network (parameters in the second equation) by applying a chunking trick, i.e., we set $D_m = D/C$ as the size of the gating vector, where $D$ is the dimension of the GNN's hidden state, and $C$ is the chunking size factor. In this setting, each gate controls $C$ neurons instead, thus drastically reducing the number of gates to be predicted.

In order to explain the practical benefits our model provided for heterophily and over-smoothing problems, we illustrate three cases in the Figure 2 and analyse them with the insights from related theoretical works. Luan et al. (2021); Ma et al. (2021) notice that heterophily is not always harmful for GNNs, and Ma et al. (2021) points out that a simple GCN can work well if similar neighbor patterns are shared across nodes with the same label, they called it "good heterophily". In Case 1, two root nodes share the same first-order neighborhood pattern, one GNN layer can capture this "good heterophily" and obtain good performance, however, a multi-layer GNN will mix in the second-order neighbor features that have different pattern, the captured "good heterophily" would be destroyed. Our model can protect the captured "good heterophily" and reject that "bad heteropihly" by learning same split points for layer 1 & 2. In Case 2, two root nodes' first-order neighbor patterns differ, but they share the same second-order neighborhood pattern. Since an ordinary GNN can only capture information *within k-hops*, they can not capture this "good heterophily" *at second-order*, whereas our model could do so with the help of split points prediction. Furthermore, we can filter out the first-order "bad heterophily".

Some works (Li et al., 2018; Chen et al., 2020a; Ma et al., 2021) show that neighbor feature smoothing is the key reason why GNN works, but too much smoothing lead to the mixing of nodes from different classes. Case 3 shows that while a GNN allows node 1 to smooth neighbor features within it's class, it also mixing node features belong to different classes for node 2. With the fine-grained ordered gating mechanism, our model can allow node 1 to aggregate similar features while filtering out information beyond node 2's own, thus helpful for homophily and over-smoothing.

The analysis of these cases shows that it is critical to predict a set of *flexible* split points. By analyzing the convergence rate of the gates, we prove that our model only constrain the relative position of the split points, the delta between split points is very flexible, i.e. the number of neurons assigned to encode the information at different orders is flexible. We provide the proof in Appendix A.2.

## 3  EXPERIMENTS

In this section, we test our model on 11 datasets. We mainly evaluate our model in 3 aspects: performances on homophily data, on heterophily data, and the robustness towards over-smoothing. We further use 2 larger datasets to see if our model could scale well w.r.t. the size of the dataset, these two datasets are from large-scale homophily and heterophily benchmarks, respectively.

**Datasets & Experiment Setup**  For homophily data, we use three citation network datasets, i.e., Cora, CiteSeer, and PubMed (Sen et al., 2008). While six heterophilous web network datasets (Pei et al., 2020; Rozemberczki et al., 2021), i.e., Actor, Texas, Cornell, Wisconsin, Squirrel and Chameleon, are used to evaluate our model on the heterophily setting. In citation networks, nodes and edges correspond to documents and citations, respectively. Node features are bag-of-words representation

Table 1: Dataset statistics.

| Dataset | Classes | Nodes | Edges | Features | $\mathcal{H}(G)$ |
|---|---|---|---|---|---|
| Cora | 7 | 2,708 | 5,429 | 1,433 | 0.81 |
| PubMed | 3 | 19,717 | 44,338 | 500 | 0.80 |
| CiteSeer | 6 | 3,327 | 4,732 | 3,703 | 0.74 |
| Cornell | 5 | 183 | 280 | 1,703 | 0.30 |
| Chameleon | 5 | 2,277 | 31,421 | 2,325 | 0.23 |
| Squirrel | 5 | 5,201 | 198,493 | 2,089 | 0.22 |
| Actor | 5 | 7,600 | 26,752 | 931 | 0.22 |
| Texas | 5 | 183 | 295 | 1,703 | 0.21 |
| Wisconsin | 5 | 251 | 466 | 1,703 | 0.11 |
| ogbn-arxiv | 40 | 169,343 | 1,166,243 | 128 | 0.66 |
| arXiv-year | 5 | 169,343 | 1,166,243 | 128 | 0.22 |

of the documents. In web networks, nodes and edges represent web pages and hyper-links, with node features being bag-of-words representation of web pages. For the over-smoothing problem, following Rong et al. (2019), we use Cora, CiteSeer, and PubMed as testbeds. For the two larger datasets, we select ogbn-arxiv from Hu et al. (2020) and arXiv-year from Lim et al. (2021). These two dataset have same nodes and edges that represent papers and citations. In ogbn-arxiv, node labels represent the papers' academic topics, which tend to be a homophily setting, while in arXiv-year, the node labels represent the publishing year, thus exhibiting more heterophily label patterns. We include the edge homophily score $\mathcal{H}(G)$ (Yan et al., 2021) for each dataset, which represents the homophily level, together with the statistics of the datasets summarized in Table 1. We use the Adam optimizer (Kingma & Ba, 2014), and apply dropout and $L_2$ regularization for $f_\theta$ and $f_\xi^{(k)}$. We also insert a LayerNorm (Ba et al., 2016) layer after every other layers. We set the hidden dimension as 256 and chunk size factor as 4. For each dataset, unless otherwise noted, a 8 layer model is used (which is already deeper than most popular GNN models). We perform a grid search to tune the hyper-parameters for all the models. More details are listed in Appendix E.

### 3.1  HOMOPHILY AND HETEROPHILY

We evaluate our model's performance on aforementioned 9 datasets, following the same data splits as Pei et al. (2020). Results are shown in Table 2. We report the mean classification accuracy with the standard deviation on the test nodes over 10 random data splits. For baselines, we include classic GNN models such as vanilla GCN (Kipf & Welling, 2016), GAT(Veličković et al., 2017), and GraphSAGE(Hamilton et al., 2017); state-of-the-art methods that are specifically designed for heterophily: H2GCN (Zhu et al., 2020b), GPRGNN (Chien et al., 2020), and GGCN (Yan et al., 2021); state-of-the-art models on three homophily citation network datasets: GCNII (Chen et al., 2020b), Geom-GCN (Pei et al., 2020). We also include models that utilize the information from

Table 2: Experimental results on homophily and heterophily datasets, for each dataset, we bold the model with the best performance.

| | Texas | Wisconsin | Actor | Squirrel | Chameleon | Cornell | CiteSeer | PubMed | Cora |
|---|---|---|---|---|---|---|---|---|---|
| **Ordered GNN** | **86.22**$_{\pm4.12}$ | **88.04**$_{\pm3.63}$ | **37.99**$_{\pm1.00}$ | **62.44**$_{\pm1.96}$ | **72.28**$_{\pm2.29}$ | **87.03**$_{\pm4.73}$ | 77.31$_{\pm1.73}$ | **90.15**$_{\pm0.38}$ | **88.37**$_{\pm0.75}$ |
| H2GCN* | 84.86$_{\pm7.23}$ | 87.65$_{\pm4.98}$ | 35.70$_{\pm1.00}$ | 36.48$_{\pm1.86}$ | 60.11$_{\pm2.15}$ | 82.70$_{\pm5.28}$ | 77.11$_{\pm1.57}$ | 89.49$_{\pm0.38}$ | 87.87$_{\pm1.20}$ |
| GPRGNN | 78.38$_{\pm4.36}$ | 82.94$_{\pm4.21}$ | 34.63$_{\pm1.22}$ | 31.61$_{\pm1.24}$ | 46.58$_{\pm1.71}$ | 80.27$_{\pm8.11}$ | 77.13$_{\pm1.67}$ | 87.54$_{\pm0.38}$ | 87.95$_{\pm1.18}$ |
| GGCN | 84.86$_{\pm4.55}$ | 86.86$_{\pm3.29}$ | 37.54$_{\pm1.56}$ | 55.17$_{\pm1.58}$ | 71.14$_{\pm1.84}$ | 85.68$_{\pm6.63}$ | 77.14$_{\pm1.45}$ | 89.15$_{\pm0.37}$ | 87.95$_{\pm1.05}$ |
| GCNII* | 77.57$_{\pm3.83}$ | 80.39$_{\pm3.4}$ | 37.44$_{\pm1.30}$ | 38.47$_{\pm1.58}$ | 63.86$_{\pm3.04}$ | 77.86$_{\pm3.79}$ | 77.33$_{\pm1.48}$ | **90.15**$_{\pm0.43}$ | **88.37**$_{\pm1.25}$ |
| Geom-GCN* | 66.76$_{\pm2.72}$ | 64.51$_{\pm3.66}$ | 31.59$_{\pm1.15}$ | 38.15$_{\pm0.92}$ | 60.00$_{\pm2.81}$ | 60.54$_{\pm3.67}$ | **78.02**$_{\pm1.15}$ | 89.95$_{\pm0.47}$ | 85.35$_{\pm1.57}$ |
| MixHop | 77.84$_{\pm7.73}$ | 75.88$_{\pm4.90}$ | 32.22$_{\pm2.34}$ | 43.80$_{\pm1.48}$ | 60.50$_{\pm2.53}$ | 73.51$_{\pm6.34}$ | 76.26$_{\pm1.33}$ | 85.31$_{\pm0.61}$ | 87.61$_{\pm0.85}$ |
| JK-Net* | 83.78$_{\pm2.21}$ | 82.55$_{\pm4.57}$ | 35.14$_{\pm1.37}$ | 45.03$_{\pm1.73}$ | 63.79$_{\pm2.27}$ | 75.68$_{\pm4.03}$ | 76.05$_{\pm1.37}$ | 88.41$_{\pm0.45}$ | 85.96$_{\pm0.83}$ |
| GCN | 55.14$_{\pm5.16}$ | 51.76$_{\pm3.06}$ | 27.32$_{\pm1.10}$ | 53.43$_{\pm2.01}$ | 64.82$_{\pm2.24}$ | 60.54$_{\pm5.30}$ | 76.50$_{\pm1.36}$ | 88.42$_{\pm0.50}$ | 86.98$_{\pm1.27}$ |
| GAT | 52.16$_{\pm6.63}$ | 49.41$_{\pm4.09}$ | 27.44$_{\pm0.89}$ | 40.72$_{\pm1.55}$ | 60.26$_{\pm2.5}$ | 61.89$_{\pm5.05}$ | 76.55$_{\pm1.23}$ | 86.33$_{\pm0.48}$ | 87.30$_{\pm1.10}$ |
| GraphSAGE | 82.43$_{\pm6.14}$ | 81.18$_{\pm5.56}$ | 34.23$_{\pm0.99}$ | 41.61$_{\pm0.74}$ | 58.73$_{\pm1.68}$ | 75.95$_{\pm5.01}$ | 76.04$_{\pm1.30}$ | 88.45$_{\pm0.50}$ | 86.90$_{\pm1.04}$ |
| MLP | 80.81$_{\pm4.75}$ | 85.29$_{\pm3.31}$ | 36.53$_{\pm0.70}$ | 28.77$_{\pm1.56}$ | 46.21$_{\pm2.99}$ | 81.89$_{\pm6.40}$ | 74.02$_{\pm1.90}$ | 87.16$_{\pm0.37}$ | 75.69$_{\pm2.00}$ |

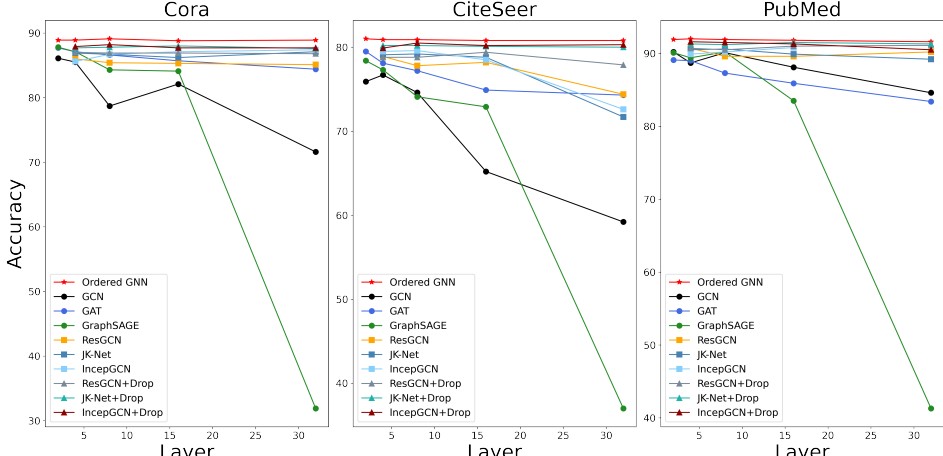

Figure 3: Node classification accuracy results for over-smoothing problem with various depths.

different hops: MixHop (Abu-El-Haija et al., 2019), JK-Net (Xu et al., 2018b); and finally a 2-layer MLP. For the baselines that have multiple architectural variants, we choose the best one for each dataset and denote the model with an asteriod "*". All the baseline results are from Yan et al. (2021), except for those on H2GCN, JK-Net and MixHop are from Zhu et al. (2020b). All of them were evaluated on the same set of data splits as ours.

Our Ordered GNN model consistently achieves SOTA both on homophilous and heterophilous datasets. On homophilous datasets, we match the performance of the SOTA model GCNII. However, GCNII performs worse on heterophilous datasets, leaving a gap of more than 20 percents on Squirrel dataset. The amazing performance on this dataset might related to the "over-squashing" problem (Alon & Yahav, 2020; Topping et al., 2021) (see Appendix B.3). We also achieve SOTA on the heterophilous datasets and consistently outperform baselines across all datasets. The superior performances on both homophily and heterophily data show the generality of our model.

## 3.2 OVER-SMOOTHING

We test our model's performance with depth 2, 4, 8, 16, 32 to evaluate the ability in alleviating the over-smoothing problem, we include following models for comparison: classical GNN models GCN (Kipf & Welling, 2016), GAT (Veličković et al., 2017) and GraphSAGE (Hamilton et al., 2017), models that modify the architecture to alleviate over-smoothing: ResGCN (Li et al., 2019), JK-Net (Xu et al., 2018b), and IncepGCN (Rong et al., 2019). We also include three state-of-the-art methods ResGCN+Drop, JK-Net+Drop, and IncepGCN+Drop that is equipped with the advanced training trick DropEdge (Rong et al., 2019) which is developed for alleviating over-smoothing. We reuse the results from Rong et al. (2019) for all baselines except GAT. For GAT, we conduct a

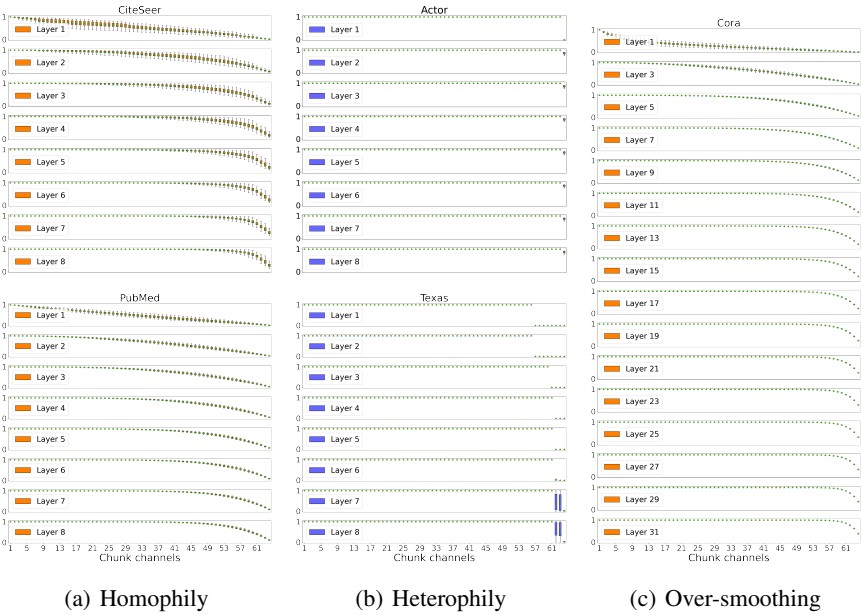

Figure 4: Visualization of gating vectors across all layers in our models on (a) homophily, (b) heterophily and (c) over-smoothing settings. Vertical and horizontal axes are gating vectors and chunk channels, respectively. Each box in the boxplot refect the distribution of the value of the corresponding gate across the whole dataset.

separate grid search for each depth to tune it's hyperparameters, with the search space the same to ours for fair comparison. Detailed configurations can be found in Appendix E. We follow the same data split from Rong et al. (2019) as used in baselines.

Performances are shown in Figure 3. Compared with the baseline models, our model consistently achieve the best performance across all datasets and all layers. Furthermore, for each dataset, our model's performance doesn't drop significantly as some baselines. Note that this is achieved without using training tricks such as DropEdge (Rong et al., 2019). This shows that our model can well alleviate the over-smoothing problem.

## 3.3 VISUALIZING THE GATING VECTORS

We checkout the gating vectors across all layers in our model, plotted in Figure 4. We can find that the gates automatically learns to behave differently for homophilous and heterophilous data, and learns to suppress message passing when needed to prevent over-smoothing.

On homophilous data the split points distributes more uniformly at first layer, with a gradually increasing trend of gating vectors. As the layer gets deeper, gates start to saturate. After layer 4, most of the positions are saturated, leaving the last very few positions open for update. This coinsides with the fact that 4 layers are the depth when many GNNs starts to show the over-smoothing phenomenon.

On heterophilous data, a large block of gates become saturated even in the first layer, emphasizes the importance of a node's ego information by suppressing most of the incoming messages. This observation is consistent with the amazing performance of a simple MLP model when deal with heterophily (see Table 2). Moreover, on Texas dataset the saturated gates keep the same for layer 1 and 2, however enlarge on layer 3, reflect the flexibility of our gating mechanism and the ability to capture "good heterophily" for high-order neighbors as we described in Section 2.5.

We also plot the gating vectors of our model on Cora dataset in Figure 4(c). We can see that almost all the gates are closed when layers become very deep. This phenomenon reveals Ordered GNN's superior performance on deep models: it manages to keep node representations distinguishable by learning to reject incoming messages from farther-away neighbors.

Table 3: Summary of classification accuracy results on larger benchmarks, we bold the model with the best performance.

| | GCN | SGC | LP | C&S | JK-Net | APPNP | GCNII | UniMP | **Ordered GNN** |
|---|---|---|---|---|---|---|---|---|---|
| ogbn-arxiv | $71.74_{\pm0.29}$ | $69.39$ | $68.50$ | $72.62$ | $72.19_{\pm0.21}$ | $66.38$ | $72.74_{\pm0.16}$ | $\mathbf{73.11}_{\pm0.21}$ | $72.81_{\pm0.21}$ |

| | GCN | SGC | LP | C&S | JK-Net | APPNP | LINK | H2GCN | **Ordered GNN** |
|---|---|---|---|---|---|---|---|---|---|
| arXiv-year | $46.02_{\pm0.26}$ | $32.83_{\pm0.13}$ | $46.07_{\pm0.15}$ | $42.17_{\pm0.27}$ | $46.28_{\pm0.29}$ | $38.15_{\pm0.26}$ | $53.97_{\pm0.18}$ | $49.09_{\pm0.10}$ | $\mathbf{54.49}_{\pm0.29}$ |

## 3.4 SCALING TO LARGER DATASETS

The larger ogbn-arxiv and arXiv-year datasets contain much more nodes, and present a complex linking pattern. We compared our model with these models: baseline graph neural models SGC (Wu et al., 2019), GCN (Kipf & Welling, 2016), label propagation based methods LP (Zhou et al., 2003), C&S (Huang et al., 2020), strong baselines: JK-Net (Xu et al., 2018b), APPNP (Klicpera et al., 2018), and state-of-the art model UniMP (Shi et al., 2020), GCNII (Chen et al., 2020b) for ogbn-arxiv, and LINK (Lim et al., 2021), H2GCN (Zhu et al., 2020b) for arXiv-year. Features, links and labels are inherited from Hu et al. (2020), all the models follow the same data split from Hu et al. (2020) and Lim et al. (2021). The baselines results are from Lim et al. (2021), Li et al. (2020a) and Huang et al. (2020).

For homophily dataset ogbn-arxiv, our model beats all the baseline methods except UniMP. This is understandable since UniMP based on the graph Transformer much larger model size, and uses the extra node label information. For heterophily dataset, our method outperform the SOTA models. The performance on these two large dataset demonstrate the scalability and expressivity of our model to capture the complex patterns in large networks.

## 3.5 ABLATION STUDY

We perform an ablation study to evaluate our proposed gating mechanism, Table 4 shows the results. "Bare GNN" denotes a simple GNN model with a mean aggregator; "+simple gating" denotes adding a gating module where each gate is predicted independently by the gating network, this is equivalent to the Gated GNN (Li et al., 2015) with a mean aggregator; "++ordered gating" denotes an ordered gating module as we introduced in Equation 6; "+++SOFTOR" indicates our proposed model Ordered GNN. We test on three settings: homophily, heterophily, and over-smoothing. We use CiteSeer for both homophily and over-smoothing settings and Wisconsin for heterophily setting. We set model layers as 8 for the first two settings and 32 for the last one.

As we can see, the "Bare GNN" model performs poorly in all three settings, especially in the heterophily setting, revealing the challenge of these settings; the "+simple gating" model and the "++ordered gating" model perform closely, both providing performance improvements, and let

Table 4: Ablation study

| Model | Homophily | Heterophily | Over-smoothing |
|---|---|---|---|
| Bare GNN | $73.79_{\pm1.57}$ | $53.92_{\pm5.79}$ | $74.70$ |
| +simple gating | $75.78_{\pm1.49}$ | $86.27_{\pm5.23}$ | $79.30$ |
| ++ordered gating | $76.45_{\pm1.35}$ | $85.69_{\pm4.90}$ | $79.40$ |
| +++SOFTOR | $77.31_{\pm1.73}$ | $88.04_{\pm3.63}$ | $80.80$ |

the model comparable to strong baselines; with the help of our proposed differentiable OR module, the "+++SOFTOR" model obtains SOTA performance, demonstrating the effectiveness of our proposed methods. Overall, the ablation study verified each component we designed for combine stage of message passing is useful and necessary.

## 4 CONCLUSION

We propose Ordered GNN, a novel GNN model that aligns the hierarchy of the rooted-tree of a central node with the ordered neurons in its node representation. This alignment implements a more organized message passing mechanism, by ordering the passed information into the node embedding w.r.t. their distance towards the node. This model results to be both more explainable and stronger in performance. It achieves the state-of-the-art on various datasets, ranging across both homophily and heterophily settings. Moreover, we show that the model could effectively prevent over-smoothing, when the model gets very deep. We discuss the connections with related works, the limitation of our model and broader impact on building more powerful GNNs in Appendix B due to the space limit.

## 5 ETHICS STATEMENT

In this work, we propose a novel GNN model to deal heterophily and over-smoothing problem. Our model integrates inductive bias from graph topology and does not contain explicit assumptions about the labels of nodes on graph, thus avoiding the occurrence of (social) bias. However, the gating mechanism of our model needs to learn from samples with labels, and it is possible to be misleading by targeting corrupted graph structure or features, which can result in negative social impact, because both internet data and social networks can be expressed with graphs. We can develop corresponding ethical guidelines, and constraint the usage of our model to avoid such problems.

## 6 REPRODUCIBILITY STATEMENT

We provide the introduction for our used datasets and model configuration in Section 3, the space of grid search to tune our model and the hardware & software experimental environment are introduced in Appendix E, we also post every set of hyper-parameters to reproduce our results in Table 7, Table 8 and Table 9, including the baseline model's. The download, load, and pre-process steps are introduced in Appendix E. We've test our model on 3 settings, the setting for heterophily & homophily are introduced in Section 3.1, the setting for over-smoothing is introduced in Section 3.2, and the setting for larger benchmarks are introduced in Section 3.4. In our released source code, we list the steps needed to reproduce results, and also list all the hyper-parameters with yaml files.

## ACKNOWLEDGEMENT

This work was sponsored by the National Natural Science Foundation of China (NSFC) grant (No. 62106143), and Shanghai Pujiang Program (No. 21PJ1405700).

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

## A  PROOF

### A.1  SOFT BINARY GATE

Ideally, we want to obtain a binary gate vector $g_v^{(k)} = (1, \ldots, 1, 0, \ldots, 0)$, and first $P_v^{(k)}$ gate is all 1s. We denote $P_v^{(k)}$ as a categorical random variable: $p(P_v^{(k)}) = \mathrm{softmax}(\ldots)$. Then for each position $l$ in the gate vector, the probability of $l$-th gate $g_v^{(k)}[l]$ being 1 could be calculated with the disjunction of $P_v^{(k)}$: the gate $g_v^{(k)}[l]$ being 1 holds when $P_v^{(k)} \geq l$, as the categories are mutually exclusive, we get the probability of $l$-th gate being 1 in gate vector: $p(g_v^{(k)}[l] = 1) = p(P_v^{(k)} \geq l) = \sum_{i=l}^{D} p(P_v^{(k)} = i)$, this can be implement with cumulative sum. Now we get the probability of a gate being 1, the gate vector can be treat as it's distribution: $\hat{g}_v^{(k)} = \mathrm{cumsum}_{\leftarrow}(\mathrm{softmax}(\ldots))$. As $g_v^{(k)}$ is a binary vector, this is equivalent to computing the expectation of it.

### A.2  CONVERGENCE RATE OF GATES

We analyze the convergence trend of gating vectors as follows: from Equation 7 we have $\tilde{g}_v^{(k)} = \tilde{g}_v^{(k-1)} + (1 - \tilde{g}_v^{(k-1)}) \circ \hat{g}_v^{(k)}$, then the finite difference for $\tilde{g}_v^{(k)}$ is $\Delta\tilde{g}_v^{(k)} = \tilde{g}_v^{(k+1)} - \tilde{g}_v^{(k)} = (1 - \tilde{g}_v^{(k)}) \circ \hat{g}_v^{(k+1)}$, since every gate's value is between 0 and 1, we have $\Delta\tilde{g}_v^{(k)} > 0$, thus the gate vectors has the trend to saturate as the layer goes deeper.

However, the rate of the gate saturation can be flexible, in order to explain this more formally, we further compare the finite difference of gates at different positions: for the gate with index $i, j$, we assume $i < j$, we ignore node $v$ notation for simplicity, then for gate $\tilde{g}_i^{(k)}, \tilde{g}_j^{(k)}$, we have $\hat{g}_i^{(k+1)} = \sum_{n=i}^{D} g_n^{*(k+1)} = \sum_{n=i}^{j-1} g_n^{*(k+1)} + \sum_{n=j}^{D} g_n^{*(k+1)} = \hat{\epsilon}^{(k+1)} + \hat{g}_j^{(k+1)}, \hat{\epsilon}^{(k+1)} \in (0, 1)$, here we denote $g_n^{*(k+1)}$ as the "raw gate" output from the linear layer of the gating network, and it's index is $n$, then we have:

$$
\begin{aligned}
&\Delta\tilde{g}_i^{(k)} - \Delta\tilde{g}_j^{(k)} \\
=&(1 - \tilde{g}_i^{(k)}) \circ (\hat{g}_i^{(k+1)}) - (1 - \tilde{g}_j^{(k)}) \circ (\hat{g}_j^{(k+1)}) \\
=&(1 - \tilde{g}_i^{(k)}) \circ (\hat{\epsilon}^{(k+1)} + \hat{g}_j^{(k+1)}) - (1 - \tilde{g}_j^{(k)}) \circ (\hat{g}_j^{(k+1)}) \\
=&\hat{g}_j^{(k+1)} \circ (\tilde{g}_j^{(k)} - \tilde{g}_i^{(k)}) + \hat{\epsilon}^{(k+1)} \circ (1 - \tilde{g}_i^{(k)})
\end{aligned}
\tag{9}
$$

here $\tilde{g}_j^{(k)} - \tilde{g}_i^{(k)} < 0$ and $1 - \tilde{g}_i^{(k)} > 0$, and $\hat{g}_j^{(k+1)} = \sum_{n=j}^{D} g_j^{*(k+1)}, \hat{\epsilon}^{(k+1)} = \sum_{n=i}^{j-1} g_j^{*(k+1)}$, since every $g_n^{*(k+1)}$ is generated by a linear layer and don't rely on historical values, then the relative size of $\hat{g}_j^{(k+1)}$ and $\hat{\epsilon}^{(k+1)}$ is not restricted, $\hat{g}_j^{(k+1)}/\hat{\epsilon}^{(k+1)} \in (0, +\infty)$, then the relative size of $\Delta\tilde{g}_i^{(k)}$ and $\Delta\tilde{g}_j^{(k)}$ is not restricted, which means the convergence rate of gates at different positions can be determined by gating network and thus leaving a *flexible* mechanism to predict $P_v^{(0)}, P_v^{(1)}, \cdots, P_v^{(K)}$: we only ensure the relative magnitude of split points, but not restrict $\Delta P_v^{(k)}$ between GNN layers, this make it possible to assign a flexible block of neurons for ego information and neighborhood information at different orders, theoretically.

## B  DISCUSSION

### B.1  CONNECTIONS WITH RELATED WORKS

**Heterophily**  Heterophily is a knotty problem for GNNs, a widely accepted method to deal with it is to allow the coefficient of message to be negative (Chien et al., 2020; Yang et al., 2021; Bo et al., 2021; Luan et al., 2021; Yan et al., 2021), this signed message let model distinguish harmful neighbor information ("bad heterophily") and reject them, we can also do this with our proposed gating mechanism; some recent works (Luan et al., 2021; Ma et al., 2021) shows that there are also

"good heterophily" that are friendly to GNNs, but didn't show a way to capture it actively rather then passively accept it, however, our model can extract it from high-order neighbors as we described in Section 2.5. It is worth noting that our model only design the combine stage, thus we can incorporate the signed message into aggregate stage to further boost the performance.

**Over-smoothing**    Over-smoothing is a very active area in graph learning, although our model focus on the combine stage of message passing mechanism, we can link it to previous successful methods and provide insight about why our model works. Similar to JK-Net (Xu et al., 2018b)'s combination of different scale information, our model allows different blocks of neurons to represent different hop information; our model locks the ego information of nodes at hop=0 in a specified block of neurons, which well preserved it and is similar to APPNP (Klicpera et al., 2018) and GCNII (Chen et al., 2020b)'s initial connection. While these useful skip connections seem to be ad-hoc, our ordered gating mechanism can be treated as a *unified* way to incorporate them; beyond just combine contexts within different hops, we further model the internal relationship (tree hierarchy) between contexts.

**Ordered Neurons**    Ordered Neurons (Shen et al., 2018b) is originally proposed to induce the syntactic tree structure from LSTM-based language models. They apply two master gating networks to control the update frequency of neurons at different positions, and employed the syntactic distance (Shen et al., 2018a) to convert the gating result into syntactic trees. Our method also relies on ordering the neurons, but the induction is in an opposite direction: the rooted-tree structure is known a priori, while the gating results are learned based on the tree structure instead.

## B.2    LIMITATIONS

As for limitations, for scale-free graphs that contain both large hubs or low-degree nodes, the rooted-tree rollout for certain nodes would vary considerably, thus our model's performance could depend on whether the data distribution is balanced, in some extreme cases, we might need to resort to the few-shot learning technique to deal with them; secondly, since our model only modifies the combined stage, we may need to modify the aggregate stage when modeling complex dependencies between nodes (e.g. knowledge graphs), we leave it for the future work.

## B.3    BROADER IMPACT

**Over-squashing**    Our model achieves amazing performance on the Squirrel datasets, and we believe that related to the "over-squashing" problem (Alon & Yahav, 2020; Topping et al., 2021). From the dataset statistics listed in Table 1, we can see that the Squirrel dataset is the densest dataset we've tested, with the ratio of edges over nodes being 40, which is 2.6 times larger than the second densest dataset Chameleon. Thus we suspect that the baselines' lousy performance is caused by the "over-squashing" problem (Alon & Yahav, 2020; Topping et al., 2021), i.e., the number of nodes contained in each node's receptive field will grow exponentially with GNN layer's growth, thus making the local information squeezed. This problem will be much more severe in more densely connected graphs. Since the baselines mainly focus on the design of the aggregate stage of the message-passing mechanism and ignore the combine stage, they cannot prevent squashing. On the other hand, our model effectively preserves the local and ego information with the ordered gating mechanism. We believe studying the over-squashing problem with our ordered form of message-passing would be an excellent future direction.

**Powerful GNNs**    Although in this work we mainly focus on our model's performance for node-level tasks, if we put it into a broader context of building more powerful GNNs (Xu et al., 2018a) that have stronger structural distinguish power, the integration of rooted-tree hierarchy can serve as an implicit learned positional encoding: with an embedding structure that forcing the order of neurons to represent the distance and importance to the central node, the central node can be aware of a precise neighborhood structure, instead of a blurred impression about it's neighbors within few hops.

This *learned* positional encoding is different from previous *injected* positional encoding (Zhang & Chen, 2018; You et al., 2019; Li et al., 2020b; Bouritsas et al., 2022) that inject pre-computed graph heuristics into node embeddings and recent *learnable* positional encoding (Dwivedi et al., 2021;

Table 5: Averaged per epoch running time on different settings.

| Model | Homophily | Heterophily | Over-smoothing | Large-scale |
|---|---|---|---|---|
| Ordered GNN | 86.52ms | 52.96ms | 243.97ms | 255.28ms |
| GAT | 102.61ms | 66.18ms | 245.21ms | 232.13ms |

Table 6: Model's performance at 64 layers when dealing with the over-smoothing.

| Methods | Cora | Citeseer | Pubmed |
|---|---|---|---|
| **Ordered GNN** | **89.10** | **80.50** | **91.60** |
| GCN | 52.00 | 44.60 | 79.70 |
| ResGCN | 79.80 | 21.20 | 87.90 |
| JKNet | 86.30 | 76.70 | 90.60 |
| IncepGCN | 85.30 | 79.00 | OOM |
| GraphSAGE | 31.90 | 16.90 | 40.70 |
| GCN+DropEdge | 53.20 | 45.60 | 79.00 |
| ResGCN+DropEdge | 84.80 | 75.30 | 90.20 |
| JKNet+DropEdge | 87.90 | 80.00 | **91.60** |
| IncepGCN+DropEdge | 88.20 | 79.90 | 90.00 |
| GraphSAGE+DropEdge | 31.90 | 25.10 | 63.20 |

Wang et al., 2022a) that provide an individual learning channel to extract the injected positional features, we actually open a door for learning positional encoding without external source. Exploring this new (implicit) positional encoding method at graph-level and link-level tasks, which are more dependent on positional and structural features, will be a very interesting future direction.

## C  RUNNING TIME

We report the averaged per epoch running times of the models on 4 settings (homophily, heterophily, over-smoothing and large-scale settings) in Table 5, we use relatively larger datasets PubMed, Actor, PubMed and ogbn-arxiv for these settings, we apply 8, 8, 32, 4 GNN layers for models respectively. We use GAT as the baseline for comparison, we implement GAT and our model with the same PyTorch Geometric framework, and share the same training/evaluation code, running on the same hardware. The results shows that our model consume similar training time compared to the popular model GAT.

## D  FURTHER INCREASING THE GNN LAYERS

Although our Ordered GNN model shows excellent performance in Table 3 when dealing with the over-smoothing problem, one may still be curious about how the model will perform at a deeper level. We showed our model's performance at 64 layers in Table 6. We can see that our model could still consistently outperform other methods on 3 datasets. Although some baselines perform well, they enjoyed the extra benefits from the advanced training trick DropEdge. Without the help of DropEdge (see ResGCN and GraphSAGE), we can see that their performance is volatile across datasets and much worse than ours; if we further compare to GCN without advanced architecture modification (e.g. residual connection in ResGCN, dense connection in JKNet), its performance deteriorated sharply.

## E  EXPERIMENTAL DETAILS

We perform a grid search to tune the hyper-parameters for models, including dropout rate, learning rate, weight decay, and MLP layers for $f_\theta$. The baseline model GAT on the over-smoothing problem shares the same search space as our model for the same configurable items. We tune dropout and

Table 7: Best hyper-parameters for homophily and heterophily setting.

| Dataset | Hyper-parameters |
|---|---|
| Cora | $dropout_\theta$:0.1, $dropout_\xi$:0.2, $L_{2_\xi}$:5e-06, $L_{2_\theta}$:5e-06, lr:0.005, MLP layers:1 |
| CiteSeer | $dropout_\theta$:0.4, $dropout_\xi$:0, $L_{2_\xi}$:0.0005, $L_{2_\theta}$:5e-08, lr:0.001, MLP layers:2 |
| PubMed | $dropout_\theta$:0.4, $dropout_\xi$:0, $L_{2_\xi}$:0.05, $L_{2_\theta}$:5e-06, lr:0.005, MLP layers:3 |
| Actor | $dropout_\theta$:0, $dropout_\xi$:0, $L_{2_\xi}$:0.0005, $L_{2_\theta}$:0.05, lr:0.01, MLP layers:2 |
| Texas | $dropout_\theta$:0.3, $dropout_\xi$:0.1, $L_{2_\xi}$:5e-06, $L_{2_\theta}$:0.05, lr:0.005, MLP layers:1 |
| Wisconsin | $dropout_\theta$:0, $dropout_\xi$:0.2, $L_{2_\xi}$:5e-06, $L_{2_\theta}$:0.05, lr:0.005, MLP layers:1 |
| Cornell | $dropout_\theta$:0.1, $dropout_\xi$:0.1, $L_{2_\xi}$:0.0005, $L_{2_\theta}$:0.05, lr:0.005, MLP layers:1 |
| Chameleon | $dropout_\theta$:0.1, $dropout_\xi$:0.1, $L_{2_\xi}$:0.0005, $L_{2_\theta}$:0.0005, lr:0.005, MLP layers:1 |
| Squirrel | $dropout_\theta$:0.3, $dropout_\xi$:0.1, $L_{2_\xi}$:0.0005, $L_{2_\theta}$:0.0005, lr:0.005, MLP layers:1 |

weight decay for $f_\theta$ and $f_\xi^{(k)}$ separately as they represent different functionalities. We run up to 2000 epochs, select the best model and apply an early stop with 200 epochs based on the accuracy of the validation set. We use NVIDIA GeForce RTX 3090 and NVIDIA GeForce RTX 2080 as the hardware environment, and use PyTorch and PyTorch Geometric as our software environment. The datasets are downloaded from PyTorch Geometric, we also use dataloader provided by PyTorch Geometric to load and pre-process datasets. We fix the random seed for reproducibility. We report a detailed search space as follows:

Search space for homophily and heterophily experiments: dropout (0,0.1,0.2,0.3,0.4), weight decay (5e-2,5e-4,5e-6,5e-8), learning rate (0.01,0.005,0.001), MLP layers (1,2,3), on PubMed and Texas datasets, we tied parameters for $f_\xi^{(k)}$ across layers; we applied empty features (Zhu et al., 2020a) and reversed edges on Chameleon and Squirrel datasets. We report the best parameters for each model-dataset combination in Table 7. Search space for over-smoothing experiments: dropout (0,0.1,0.2,0.3,0.4,0.5), weight decay (5e-1,5e-2,5e-4,5e-6,5e-8), learning rate (0.01,0.005,0.001), MLP layers (1,2). We tied parameters for $f_\xi^{(k)}$ across layers for all models except models with 32 layers. We report the best parameters for each model-dataset combination in Table 8. Search space for expressivity experiments: dropout (0.05,0.1,0.15,0.2), weight decay (0,5e-4,5e-8), we fix the MLP layers as 3, and learning rate as 0.005, we tied parameters for $f_\xi^{(k)}$ across layers on ogbn-arxiv dataset. We report the best parameters for each model-dataset combination in Table 9.

Table 8: Best hyper-parameters for over-smoothing problems.

| Dataset | Method | Layers | Hyper-parameters |
|---|---|---|---|
| Cora | Ordered GNN | 2 | $dropout_\theta$:0.4, $dropout_\xi$:0.4, $L_{2_\xi}$:5e-08, $L_{2_\theta}$:0.0005, lr:0.01, MLP layers:1 |
| | | 4 | $dropout_\theta$:0.4, $dropout_\xi$:0.2, $L_{2_\xi}$:5e-06, $L_{2_\theta}$:5e-08, lr:0.001, MLP layers:2 |
| | | 8 | $dropout_\theta$:0.1, $dropout_\xi$:0.3, $L_{2_\xi}$:5e-08, $L_{2_\theta}$:0.0005, lr:0.01, MLP layers:2 |
| | | 16 | $dropout_\theta$:0.2, $dropout_\xi$:0.1, $L_{2_\xi}$:5e-06, $L_{2_\theta}$:0.05, lr:0.005, MLP layers:2 |
| | | 32 | $dropout_\theta$:0.5, $dropout_\xi$:0, $L_{2_\xi}$:0.05, $L_{2_\theta}$:0.05, lr:0.001, MLP layers:2 |
| | GAT | 2 | $dropout$:0.4, $L_2$:0.0005, lr:0.005 |
| | | 4 | $dropout$:0.5, $L_2$:5e-08, lr:0.005 |
| | | 8 | $dropout$:0.4, $L_2$:5e-06, lr:0.01 |
| | | 16 | $dropout$:0.3, $L_2$:0, lr:0.001 |
| | | 32 | $dropout$:0.3, $L_2$:5e-08, lr:0.001 |
| CiteSeer | Ordered GNN | 2 | $dropout_\theta$:0.1, $dropout_\xi$:0.3, $L_{2_\xi}$:5e-06, $L_{2_\theta}$:0.05, lr:0.005, MLP layers:1 |
| | | 4 | $dropout_\theta$:0.3, $dropout_\xi$:0.2, $L_{2_\xi}$:5e-06, $L_{2_\theta}$:0.0005, lr:0.01, MLP layers:1 |
| | | 8 | $dropout_\theta$:0.4, $dropout_\xi$:0, $L_{2_\xi}$:5e-06, $L_{2_\theta}$:5e-06, lr:0.001, MLP layers:2 |
| | | 16 | $dropout_\theta$:0.4, $dropout_\xi$:0, $L_{2_\xi}$:0.05, $L_{2_\theta}$:5e-06, lr:0.001, MLP layers:2 |
| | | 32 | $dropout_\theta$:0.5, $dropout_\xi$:0, $L_{2_\xi}$:0.0005, $L_{2_\theta}$:5e-08, lr:0.001, MLP layers:2 |
| | GAT | 2 | $dropout$:0.5, $L_2$:5e-06, lr:0.005 |
| | | 4 | $dropout$:0.5, $L_2$:5e-06, lr:0.001 |
| | | 8 | $dropout$:0, $L_2$:5e-06, lr:0.005 |
| | | 16 | $dropout$:0.2, $L_2$:5e-06, lr:0.001 |
| | | 32 | $dropout$:0.1, $L_2$:5e-08, lr:0.001 |
| PubMed | Ordered GNN | 2 | $dropout_\theta$:0.3, $dropout_\xi$:0.2, $L_{2_\xi}$:5e-06, $L_{2_\theta}$:0.0005, lr:0.01, MLP layers:2 |
| | | 4 | $dropout_\theta$:0.2, $dropout_\xi$:0.1, $L_{2_\xi}$:0.0005, $L_{2_\theta}$:0.0005, lr:0.01, MLP layers:2 |
| | | 8 | $dropout_\theta$:0.2, $dropout_\xi$:0.1, $L_{2_\xi}$:5e-08, $L_{2_\theta}$:5e-06, lr:0.005, MLP layers:2 |
| | | 16 | $dropout_\theta$:0.2, $dropout_\xi$:0.1, $L_{2_\xi}$:0.05, $L_{2_\theta}$:0.05, lr:0.001, MLP layers:2 |
| | | 32 | $dropout_\theta$:0.3, $dropout_\xi$:0, $L_{2_\xi}$:0.5, $L_{2_\theta}$:0.0005, lr:0.005, MLP layers:2 |
| | GAT | 2 | $dropout$:0.2, $L_2$:5e-08, lr:0.01 |
| | | 4 | $dropout$:0.1, $L_2$:0, lr:0.01 |
| | | 8 | $dropout$:0.2, $L_2$:5e-08, lr:0.01 |
| | | 16 | $dropout$:0, $L_2$:5e-06, lr:0.005 |
| | | 32 | $dropout$:0, $L_2$:5e-08, lr:0.001 |

Table 9: Best hyper-parameters on larger datasets.

| Dataset | Hyper-parameters |
|---|---|
| ogbn-arxiv | $dropout_\theta$:0.2, $dropout_\xi$:0.05, $L_{2_\xi}$:0.0005, $L_{2_\theta}$:5e-08 |
| arXiv-year | $dropout_\theta$:0.1, $dropout_\xi$:0.05, $L_{2_\xi}$:5e-08, $L_{2_\theta}$:0 |

