# OpenReview forum: "Ordered GNN: Ordering Message Passing to Deal with Heterophily and Over-smoothing"
_ICLR.cc/2023/Conference — ICLR 2023 poster_

### Official Review · Reviewer_gn7g · 2022-10-22

**Confidence:** 3
**Correctness:** 4
**Technical Novelty And Significance:** 3
**Empirical Novelty And Significance:** 3
**Recommendation:** 6

**Clarity, Quality, Novelty And Reproducibility:**

This paper develops a new method for message passing. It aligns the hierarchy of the rooted-tree of a central node with the ordered neurons. The motivation is clear and the paper is well-organized. The experiments are also comprehensive to evaluate the effect of each component.


**Strength And Weaknesses:**

The strength of this paper includes:
1. The paper is well-motivated.
2. The experiments are extensive.
3. The analysis is sufficient.
 The weaknesses are:
1. The experimental results are not impressive. The improvement is not significant.
2. The comparison methods are not STOA. Some recent works are ignored.

**Summary Of The Paper:**

This paper proposes a new message passing method for GNN. It orders the message passing into the node representation with specific blocks of neurons targeted for message passing within specific hops. Extensive experiments show the effectiveness of the proposed method on homophily and heterophily data. It also alleviates the over-smoothing problem.

**Summary Of The Review:**

My worry about this paper is on the experimental part. Most of the comparison methods are published in or before 2020. Only GGCN, which was published in 2021, is compared. Much progress has been achieved in 2021 and 2022. Therefore, it is fairer to compare some of them. Furthermore, the improvement is also not significant in most cases.

---

> ### Author Response · Authors · 2022-11-10
> **Response to Reviewer gn7g**
>
> > The experimental results are not impressive. The improvement is not significant.
>
> > The comparison methods are not STOA. Some recent works are ignored.
>
> > My worry about this paper is on the experimental part. Most of the comparison methods are published in or before 2020. Only GGCN, which was published in 2021, is compared. Much progress has been achieved in 2021 and 2022. Therefore, it is fairer to compare some of them. Furthermore, the improvement is also not significant in most cases.
>
> (Merged response) To alleviate your concerns on the experimental part, we found a recent paper [1] accepted in NeurIPS 2022, which follows the same experimental setting and dataset splits as ours. We directly append the results of our model to Table 1 of the paper [1], please refer to Response to Reviewer UBsM (Part 3/3) for the table. We can see that our model beat all the baselines on 7/9 datasets. Furthermore, we beat all the 3 models (including variants) proposed by paper [1] on 8/9 datasets, which shows the superiority of our model. Specifically, on dataset Squirrel, we beat the best model (O(d)-NSD) in Table 1 of the paper [1] by 6 points, which is a considerable margin, considering we only modify the combine stage of message passing, we believe our design is significant in improving GNN’s performance.
>
> [1] Neural Sheaf Diffusion: A Topological Perspective on Heterophily and Oversmoothing in GNNs. NeurIPS 2022

---

### Official Review · Reviewer_44qL · 2022-10-24

**Confidence:** 5
**Correctness:** 3
**Technical Novelty And Significance:** 2
**Empirical Novelty And Significance:** 3
**Recommendation:** 5

**Clarity, Quality, Novelty And Reproducibility:**

The motivation of the proposed method is kind of unclear and lack some evaluations.

**Strength And Weaknesses:**

### Strengths:
It is interesting to introduce the ordered neurons into graph representation learning, and it is one straightforward solution for the over-smoothing problem. The experimental results demonstrate the effectiveness in preventing this problem.

### Weakness:

1. The motivation for separating the different hops is not clear. As mentioned in the introduction, what are the advantages of “model the information exactly at some orders”?

2. The evaluations on the node-wise split points are lacking. Designing the personalized operation for each node helps to improve the model performance [1,2]. It seems that different nodes have different split points as mentioned in Section 2.3. This aspect is not evaluated in the experiments, either by comparing with some node-wise GNNs or showing the different split points. Besides, there lacks a detailed discussion about the ordered neuron methods to verify the contributions of this paper.

3. The information extraction on the ego features. The self-loop-free graphs are utilized in this paper. How to incorporate the node features from itself?

### Refs:
[1] Graph Neural Networks with Node-wise Architecture. KDD 2022

[2] Policy-GNN: Aggregation Optimization for Graph Neural Networks. KDD 2020


**Summary Of The Paper:**

This paper proposed one ordered GNNs to mix the features calculated from different hops. This method can achieve the SOTA performance on both homophily and heterophily graphs, and could prevent the over-smoothing problem.

**Summary Of The Review:**

It is interesting to employ the ordered neurons in the graph representation learning and the experiments can demonstrate the effectiveness. However, the proposed method is not well motivated and some baselines are lacked.

---

> ### Author Response · Authors · 2022-11-10
> **Response to Reviewer 44qL**
>
> > The motivation for separating the different hops is not clear. As mentioned in the introduction, what are the advantages of “model the information exactly at some orders”?
>
> The motivation to separate the information of different hops is to avoid mixing neighbor information between different hops. In the Introduction, we introduced several early works that tried to make this separation and pointed out their problems. We conducted a case study in Section 2.5 to help understand the insight and practical benefits.
>
> Modeling the information strictly at some orders allows the model to capture "good heterophily" when dealing with heterophily and allows nodes to easily preserve their ego and local information when facing the over-smoothing problem. We explain this in detail through a case study in Section 2.5. The excellent performance of our model in Sections 3.1 and 3.2 also verifies the effectiveness of this operation.
>
> > The evaluations on the node-wise split points are lacking. Designing the personalized operation for each node helps to improve the model performance [1,2]. It seems that different nodes have different split points as mentioned in Section 2.3. This aspect is not evaluated in the experiments, either by comparing with some node-wise GNNs or showing the different split points. Besides, there lacks a detailed discussion about the ordered neuron methods to verify the contributions of this paper.
>
> It is worth noting that in Table 2, the model GGCN [1] is a node-wise model, which has two personalized operations for each node: degree correction and signed messages, and we outperform it on all datasets. Following your suggestion, we’ve checked baseline [2] and baseline [3]. Baseline [2] uses NAS to search GNN architectures, and baseline [3] uses reinforcement learning to design GNN aggregation strategies. Although the dataset they use coincides with ours, unfortunately neither of them provides source code, best hyperparameters, or trained model, so we are unable to reproduce on our side. We’ll put these two baselines in our Introduction.
>
> To further alleviate your doubts, we found a recent paper [4] accepted in NeurIPS 2022, which follows the same experimental setting and dataset splits as ours. We directly append the results of our model to Table 1 of the paper [4], please refer to Response to Reviewer UBsM (Part 3/3) for the table. We can see that our model beat all the baselines on 7/9 datasets. In these baselines, FAGCN [5] and GGCN [1] have designed personalized operations for each node. As for the verification of different node partitions, we showed the visualization of the gate vector in Figure 4, and we can see that different nodes do have personalized gates.
>
> As for the discussion of Ordered Neurons [6], we’ve discussed the relationship between our ordered gating mechanism and split points in Appendix A.1, and analyzed the convergence rate of ordered gates in Appendix A.2. In addition, we’ve explained the relationship between our method and Ordered Neurons [6] in Appendix B.1: our method also relies on ordering the neurons, but the induction is in the opposite direction: the rooted-tree structure is known a priori, while the gating results are learned based on the tree structure instead.
>
> > The information extraction on the ego features. The self-loop-free graphs are utilized in this paper. How to incorporate the node features from itself?
>
> In Section 2.2, we’ve explained that $\mathcal{T}_{v}^{(0)}$ corresponds to the neurons used to represent the node’s ego information, which means that for the first time of message passing, the node embedding is split into two blocks, one for node features from itself and another one for aggregated messages.
>
> [1] Two Sides of the Same Coin: Heterophily and Oversmoothing in Graph Convolutional Neural Networks. ICDM 2022
>
> [2] Graph Neural Networks with Node-wise Architecture. KDD 2022
>
> [3] Policy-GNN: Aggregation Optimization for Graph Neural Networks. KDD 2020
>
> [4] Neural Sheaf Diffusion: A Topological Perspective on Heterophily and Oversmoothing in GNNs. NeurIPS 2022
>
> [5] Beyond Low-frequency Information in Graph Convolutional Networks. AAAI 2021
>
> [6] Ordered Neurons: Integrating Tree Structures into Recurrent Neural Networks. ICLR 2019

---

### Official Review · Reviewer_rLLb · 2022-11-02

**Confidence:** 3
**Correctness:** 3
**Technical Novelty And Significance:** 3
**Empirical Novelty And Significance:** 3
**Recommendation:** 8

**Clarity, Quality, Novelty And Reproducibility:**

The presentation is clear. The experiment result is convincing. But, the source code is not released which prevents me from the judgment of the reproducibility.

Although the ordered neuron trick was introduced in prior work, the author adapted it to literately preserving the ego info which is still considered as a good contribution in terms of novelty.

**Strength And Weaknesses:**

Strength:
1. Intuitive and effective design of the gating function to iteratively update the feature while preserving the ego node info by inductive bias using ordered neurons.
2. Sufficient number of experiment in both small and large scale dataset.
3. Good presentation, easy to follow.

To be clear:
1. For the experiment result in the Squirrel dataset, the performance surpass the baselines by a quite large margin. I'm wondering if there is an underline reason for such improvement.
2. Fig 3 shows some baselines which previously designed to deal with the over-smoothing problem are actually pretty good. Even though there overall performance doesn't beat the proposed Ordered GNN. A good question is that if the number of layers is further increased, what would happen? In another word, is such carefully designed gating procedure actually needed for solving the over-smoothing problem?


**Summary Of The Paper:**

This paper presents an ordering mechanism for the messages passing in GNN for node representation learning. In particular they are focused on the combining stage in the GNN procedure. The ordering mechanism is achieved by the rooted-tree hierarchy and a curated gating function. which can iteratively preserve the ego information and gradually absorb k-hop neighboring features. The authors claimed to address the problem of “over-smoothing” and “heterophily”. The experiment results show the performance boost in both settings above. And the large scale dataset experiment demonstrated the scalability of the proposed method.

**Summary Of The Review:**

The proposed Ordered GNN approach is clear and the experiments and the ablation studies demonstrated the effectiveness of each part of the design. The algorithm is tested in both small and large scale dataset. I recommend for accept.

---

> ### Author Response · Authors · 2022-11-10
> **Response to Reviewer rLLb**
>
> > For the experiment result in the Squirrel dataset, the performance surpass the baselines by a quite large margin. I'm wondering if there is an underline reason for such improvement.
>
> From the dataset statistics listed in Table 1, we can see that the Squirrel dataset is the densest dataset we've tested, with the ratio of *# edges* over *# nodes* being ~40, which is ~2.6 times larger than the second densest dataset Chameleon. We suspect that the baselines' lousy performance is caused by the "over-squashing" problem [1,2], i.e., the number of nodes contained in each node's receptive field will grow exponentially with GNN layer's growth, thus making the local information squeezed. This problem will be much more severe in more densely connected graphs. Since the baselines mainly focus on the design of the aggregate stage of the message-passing mechanism and ignore the combine stage, they cannot prevent squashing. On the other hand, our model effectively preserves the local and ego information with the ordered gating mechanism. Note that this work mainly focuses on the heterophily and over-smoothing problem, but we believe studying the over-squashing problem with our ordered form of message-passing would be an excellent future direction.
>
> > Fig 3 shows some baselines which previously designed to deal with the over-smoothing problem are actually pretty good. Even though there overall performance doesn't beat the proposed Ordered GNN. A good question is that if the number of layers is further increased, what would happen? In another word, is such carefully designed gating procedure actually needed for solving the over-smoothing problem?
>
> Following your suggestion, we've tested our model's performance with 64 layers, which is really deep for a GNN model. We report the results together with baselines as follows.
>
> | Methods | Cora | Citeseer | Pubmed |
> | ---- | ---- | ---- | ---- |
> |**Ordered GNN** | **89.10** | **80.50** | **91.60** |
> | GCN | 52.00 | 44.60 | 79.70 |
> | ResGCN | 79.80 | 21.20 | 87.90 |
> | JKNet | 86.30 | 76.70 | 90.60 |
> | IncepGCN | 85.30 | 79.00 | OOM |
> | GraphSAGE | 31.90 | 16.90 | 40.70 |
> | GCN+DropEdge | 53.20 | 45.60 | 79.00 |
> | ResGCN+DropEdge | 84.80 | 75.30 | 90.20 |
> | JKNet+DropEdge | 87.90 | 80.00 | **91.60** |
> | IncepGCN+DropEdge | 88.20 | 79.90 | 90.00 |
> | GraphSAGE+DropEdge | 31.90 | 25.10 | 63.20 |
>
> We can see that our model could still consistently beat other methods on 3 datasets. Although some baselines perform well, they enjoyed the extra benefits from the advanced training trick DropEdge. Without the help of DropEdge (see ResGCN and GraphSAGE), we can see that their performance is volatile across datasets and much worse than ours; if we further compare to GCN without advanced architecture modification (e.g. residual connection in ResGCN, dense connection in JKNet), its performance deteriorated sharply.
>
> From the results above, JKNet and it's variant JKNet+DropEdge seems good at dealing with the over-smoothing problem, it performs more robust on 3 datasets compared to other baselines. However, if we check it's performance in dealing with the heterophily problem (see Table 2 of our paper), we can see that it's performance is unstable, and leaves a gap of 18 points to our model on Chameleon dataset. This reveals the designs introduced to deal with over-smoothing previously may not be suitable for other problems, while our proposed method consistently performs well on two problems. Considering that we've only modified one component (combine stage) of the message-passing mechanism, then the GNNs' performances across all datasets are dramatically improved and become robust to very deep layers, we believe our proposed method hits the core of the over-smoothing problem in a better way. In addition, our approach is also orthogonal to those tricks used in the baselines. Thus, our model could potentially perform even better in dealing with over-smoothing problem if combined with those bells and whistles.
>
> > But, the source code is not released which prevents me from the judgment of the reproducibility.
>
> We’ve released the anonymous links in the comments, including the source code, splitting files, experimental steps, and the hyperparameter configurations to reproduce the results.
>
> [1] On the bottleneck of graph neural networks and its practical implications. ICLR 2021
>
> [2] Understanding over-squashing and bottlenecks on graphs via curvature. ICLR 2022

---

### Official Review · Reviewer_UBsM · 2022-11-05

**Confidence:** 4
**Clarity, Quality, Novelty And Reproducibility:** Please refer to the Strength And Weak…
**Correctness:** 3
**Technical Novelty And Significance:** 2
**Empirical Novelty And Significance:** 2
**Recommendation:** 3

**Strength And Weaknesses:**

This paper presents an Ordered GNN for handling the heterophily problem, which attempts to improve the updating process in the message passing scheme. The paper studies an important problem, i.e., how to extend current GNNs to heterophilic graphs. Unfortunately, there are many confusions and presentation problems in this paper, which makes it hard to follow. Besides, I also have several concerns about its motivation, detailed method, and evaluations.

Comments
1. I am afraid that there are many confusions and grammar errors in the paper, which makes it hard to follow. For example, in Sec. 2.2, I am wondering what "one round of message passing", "the neurons used to represent the nodes’ ego information", "the neurons used to represent message passing within Tao^{(k-1)} should be a subset of those for Tao^{(k)}", and "the delta between k-1 and k" mean. Besides, the paper may benefit from revising certain presentations, e.g., "its k-th order rooted tree", "And since …", "Apparently here we have…", ". where Tao^{(0)} corresponds to…". I am afraid that the paper needs to be carefully revised.
2. I am afraid that some utilized mathematic symbols lack formal definitions and thus induce unnecessary confusions. For example, Tao^{(k)}_{v} is defined as the "k-th order rooted tree" of node v. However, it is not clear what its subset operation is (as shown in Eq. (3)) and why "Tao^{(0)}_v corresponds to the neurons used to represent the ego representations".
3. I am afraid that the reviews of related work in Sec. 1 are confusing. This paper focuses on two drawbacks of the message passing scheme, i.e., over-smoothing and heterophily. It is not clear which problems these reviewed literatures were designed to solve. It actually seems that this paper considers these two problems as the same problem.
4. In Sec. 2.5, I am afraid that the intuitive analysis for Fig. 2 is not convincing. The paper states "In Case 1, two root nodes share the same first-order neighborhood pattern, one GNN layer can capture this good heterophily and obtain good performance". This analysis only considers node 1 and node 2. However, the children of these two nodes possess different situations which are not considered in this analysis. The analysis for Case 2 also has a similar problem.
5. I am afraid that the results of the employed baselines in Table 2 are most borrowed from a preprint paper, which is not convincing. For example, I am just wondering why the results of GRP-GNN are much lower than its reported ones.
6. I am just wondering how the proposed method can specifically solve the over-smoothing and heterophily problems.


**Summary Of The Paper:**

This paper presents an Ordered GNN for handling the heterophily problem, which attempts to improve the updating process in the message passing scheme. The paper studies an important problem, i.e., how to extend current GNNs to heterophilic graphs.

**Summary Of The Review:**

The paper studies an important problem, i.e., how to extend current GNNs to heterophilic graphs. Unfortunately, there are many confusions and presentation problems in this paper, which makes it hard to follow. Besides, I also have several concerns about its motivation, detailed method, and evaluations.

---

> ### Author Response · Authors · 2022-11-10
> **Response to Reviewer UBsM (Part 1/3)**
>
> > I am afraid that there are many confusions and grammar errors in the paper, which makes it hard to follow. For example, in Sec. 2.2, I am wondering what "one round of message passing", "the neurons used to represent the nodes’ ego information", "the neurons used to represent message passing within Tao^{(k-1)} should be a subset of those for Tao^{(k)}", and "the delta between k-1 and k" mean. Besides, the paper may benefit from revising certain presentations, e.g., "its k-th order rooted tree", "And since …", "Apparently here we have…", ". where Tao^{(0)} corresponds to…". I am afraid that the paper needs to be carefully revised.
>
> "one round of message passing" means one time of message passing on the graph; "the neurons used to represent the nodes' ego information" means a set of dimensions in node embedding to represent the nodes' ego information, "neurons" here indicate nodes (or hidden units, neural nodes, etc.) in the neural network; "the neurons used to represent message passing within Tao^{(k-1)} should be a subset of those for Tao^{(k)}" means the set of dimensions in node embedding to represent message passing within $\mathcal{T}\_{v}^{(k-1)}$ should be a subset of the set of dimensions in node embedding for representing message passing within $\mathcal{T}\_{v}^{(k)}$, we've shortened the sentence in our paper for better readability; "the delta between k-1 and k" means the difference of information between k and k-1 hop, which reflects the neighbor information strictly at k-th order.
>
> The presentations like “its k-th order rooted tree”, “And since …”, “Apparently here we have…”, “. where Tao^{(0)} corresponds to…” are commonly used conjunctions and can be understood if read the context of that sentence, we’ve also plotted the rooted-tree and concepts around it in Figure 1 to help to understand, we’ve mentioned Figure 1 in Introduction and the first paragraph of Section 2.2; considering the comments to our presentation from reviewers, e.g., “Good presentation, easy to follow.” (from Reviewer rLLb), we believe this paper might not be that hard to follow. However, we are ready to answer any of your further questions about our presentation.
>
> > I am afraid that some utilized mathematic symbols lack formal definitions and thus induce unnecessary confusions. For example, Tao^{(k)}_{v} is defined as the "k-th order rooted tree" of node v. However, it is not clear what its subset operation is (as shown in Eq. (3)) and why "Tao^{(0)}_v corresponds to the neurons used to represent the ego representations".
>
> The formal definition of $\mathcal{T}\_{v}^{(k)}$ in our paper is “For node v, its k-th order rooted-tree $\mathcal{T}\_{v}^{(k)}$ is composed by putting its first order neighbors as its children, and then recursively for each child, putting the neighboring nodes of the child as its children (except the child that is also the ancester of the node), unitl its k-th order neighbors (Figure 1).”, which is a clear definition and can be understood if refer to Figure 1; before we show Eq. (3), we’ve explained the relationship between subtrees: “the $k-1$-th rooted-tree $\mathcal{T}\_{v}^{(k-1)}$ is a subtree of the $k$-th rooted-tree  $\mathcal{T}\_{v}^{(k)}$, with the same root $v$ and the same tree structure up to depth $k-1$”, if read Eq. (3) with these contexts, the subset operation and “Tao^{(0)}_v corresponds to the neurons used to represent the ego representations” can be understood.
>
> > I am afraid that the reviews of related work in Sec. 1 are confusing. This paper focuses on two drawbacks of the message passing scheme, i.e., over-smoothing and heterophily. It is not clear which problems these reviewed literatures were designed to solve. It actually seems that this paper considers these two problems as the same problem.
>
> In this paper, we focus on improving the message-passing mechanism. Thus the related works are organized based on which aspect of message-passing mechanism they focus on and we divide previous works into 3 parts: works focus on the aggregate stage, works focus on the integration of messages from different hops, and works focus on the data side of message passing. A previous work [1] working on heterophily and over-smoothing problems also organizes their related works by merging the works to solve these 2 problems (see the second and third paragraph of Section 1 in paper [1]).
> We didn’t consider these two problems as the same problem:
> - In the Introduction, the sources of the two problems are discussed separately.
> - In the case study part, we proposed cases for two problems and offered analysis separately.
> - In the experiment part, we separated the evaluation of two problems in Section 3.1 and Section 3.2.
> - In the discussion part (Appendix B.1), we discussed our model’s relation with previous works separately.

---

> > ### Author Response · Authors · 2022-11-10
> > **Response to Reviewer UBsM (Part 2/3)**
> >
> > > In Sec. 2.5, I am afraid that the intuitive analysis for Fig. 2 is not convincing. The paper states "In Case 1, two root nodes share the same first-order neighborhood pattern, one GNN layer can capture this good heterophily and obtain good performance". This analysis only considers node 1 and node 2. However, the children of these two nodes possess different situations which are not considered in this analysis. The analysis for Case 2 also has a similar problem.
> >
> > Our case study part has followed the widely accepted manner in graph machine learning. We list several of them here:
> > - in Figure 2 and Figure 3 of the paper [2], they proposed cases of rooted trees to explain the difference between aggregator designs. They didn’t analyze the children of root nodes.
> > - In Figure 1 of the paper [3], they proposed a case of neighborhood structure of a central node to explain the heterophily problem, also didn’t analyze the children of the central node.
> > - In Figure 1 of paper [4], they proposed cases of rooted subtrees to discuss the non-isomorphic problem, and only analyze one node on 2 graphs at a special position.
> > - In Figure 2 of paper [5], they proposed cases of rooted-trees to explain the “good heterophily”, and also only analyze the root nodes.
> >
> > > I am afraid that the results of the employed baselines in Table 2 are most borrowed from a preprint paper, which is not convincing. For example, I am just wondering why the results of GRP-GNN are much lower than its reported ones.
> >
> > We follow the experimental settings in the paper [3], because it is usually considered the first paper to consider the heterophily problem explicitly. We borrowed results from the paper [6] because it’s one of the best performance models following the same setting as the paper [3]’s. It has been submitted to ICLR 2022 and accepted in ICDM 2022 recently. It has a citation of more than 50, which can be treated as an important paper that can not be ignored. Furthermore, a recently accepted paper [7] in NeurIPS 2022 also borrowed results from the paper [6] and followed the settings in the paper [3]. We directly append the results of our models to Table 1 of paper [7] and show them as follows. We can see that our model beat all the baselines on 7/9 datasets.
> >
> > To address your concern about GPR-GNN [8], we’ve checked the experimental settings in their paper. In the first paragraph of Appendix A.9, they said, “For the training set, we ensure that number of nodes from each class is approximately the same and keep the total number of training nodes close to 2.5%/60%. **For the validation set, we randomly sample 2.5%/20% of the nodes and place the remaining ones into the test set**”, which means they generate their dataset splitting. They do not follow the splitting in paper [3], paper [6], and paper [7] . The dataset splitting in node classification task is critically sensitive because the graph structure links the labeled and unlabeled examples. This might be the reason GPR-GNN performs differently, as reported in the paper [6].
> >
> > > I am just wondering how the proposed method can specifically solve the over-smoothing and heterophily problems.
> >
> > We’ve discussed the connections with related works in Appendix B.1, and linking our model to previous successful methods, it can help understanding how our model works; we’ve put the case study and linked to the related theoretical works in Section 2.5 to explain the practical benefits our model provided for heterophily and over-smoothing problems, we’ve further provided the proof for the flexible split points prediction in Appendix A.2, this verifies the ability of our model to handle these two problems. The visualizations in Section 3.3 reveal how our model makes the decision for each node under different neighbor label patterns. This offers explainability to our model’s behavior and can further help us understand how our proposed method can specifically solve the over-smoothing and heterophily problems.

---

> > > ### Author Response · Authors · 2022-11-10
> > > **Response to Reviewer UBsM (Part 3/3)**
> > >
> > > | Methods | Texas | Wisconsin | Film | Squirrel | Chameleon | Cornell | Citeseer | Pubmed | Cora |
> > > | ---- | ---- | ---- | ---- | ---- | ---- | ---- | ---- | ---- | ---- |
> > > |**Ordered GNN** | $\textbf{86.22}{\pm4.12}$ | ${88.04}{\pm3.63}$ | $\textbf{37.99}{\pm1.00}$ | $\textbf{62.44}{\pm1.96}$ | $\textbf{72.28}{\pm2.29}$ | $\textbf{87.03}{\pm4.73}$ | $77.31{\pm1.73}$ | $\textbf{90.15}{\pm0.38}$ | $\textbf{88.37}{\pm0.75}$ |
> > > |**Diag-NSD** | ${85.67} { \pm 6.95}$ | ${88.63} { \pm 2.75}$ | ${37.79} { \pm 1.01}$ | ${54.78} { \pm 1.81}$ | ${68.68} { \pm 1.73}$ | ${86.49} { \pm 7.35}$ | ${77.14} { \pm 1.85}$ | $89.42 { \pm 0.43}$ | $87.14 { \pm 1.06}$|
> > > |**O(d)-NSD** | ${85.95} { \pm 5.51}$ | $\textbf{89.41} { \pm 4.74}$ | ${37.81} { \pm 1.15}$ | ${56.34} { \pm 1.32}$  | ${68.04} { \pm 1.58}$ | ${84.86} { \pm 4.71}$ | $76.70 { \pm 1.57}$ | ${89.49} { \pm 0.40}$ | $86.90 { \pm 1.13}$ |
> > > |**Gen-NSD** | $82.97 { \pm 5.13}$ | ${89.21} { \pm 3.84}$ | ${37.80} { \pm 1.22}$ | $53.17 { \pm 1.31}$ | $67.93 { \pm 1.58}$ | ${85.68} { \pm 6.51}$ | $76.32 { \pm 1.65}$ | $89.33 { \pm 0.35}$ | $87.30 { \pm 1.15}$ |
> > > |GGCN | ${84.86} { \pm 4.55}$ | $86.86 { \pm 3.29}$ | $37.54 { \pm 1.56}$ | ${55.17} { \pm 1.58}$ | ${71.14} { \pm 1.84}$ | ${85.68} { \pm 6.63}$ | ${77.14} { \pm 1.45}$ | $89.15 { \pm 0.37}$ | ${87.95} { \pm 1.05}$ |
> > > |H2GCN | ${84.86} { \pm 7.23}$ | $87.65 { \pm 4.98}$ | $35.70 { \pm 1.00}$ | $36.48 { \pm 1.86}$ | $60.11 { \pm 2.15}$ | $82.70 { \pm 5.28}$ | $77.11 { \pm 1.57}$ | ${89.49} { \pm 0.38}$ | ${87.87} { \pm 1.20}$ |
> > > |GPRGNN | $78.38 { \pm 4.36}$ | $82.94 { \pm 4.21}$ | $34.63 { \pm 1.22}$ | $31.61 { \pm 1.24}$ | $46.58 { \pm 1.71}$ | $80.27 { \pm 8.11}$ | $77.13 { \pm 1.67}$ | $87.54 { \pm 0.38}$ | ${87.95} { \pm 1.18}$ |
> > > |FAGCN | $82.43 { \pm 6.89}$ | $82.94 { \pm 7.95}$ | $34.87 { \pm 1.25}$ | $42.59 { \pm 0.79}$ | $55.22 { \pm 3.19}$ | $79.19 { \pm 9.79}$ | N/A | N/A | N/A |
> > > |MixHop | $77.84 { \pm 7.73}$ | $75.88 { \pm 4.90}$ | $32.22 { \pm 2.34}$ | $43.80 { \pm 1.48}$ | $60.50 { \pm 2.53}$ | $73.51 { \pm 6.34}$ | $76.26 { \pm 1.33}$ | $85.31 { \pm 0.61}$ | $87.61 { \pm 0.85}$ |
> > > |GCNII | $77.57 { \pm 3.83}$ | $80.39 { \pm 3.40}$  | $37.44 { \pm 1.30}$ | $38.47 { \pm 1.58}$ | $63.86 { \pm 3.04}$ | $77.86 { \pm 3.79}$ | ${77.33} { \pm 1.48}$ | $\textbf{90.15} { \pm 0.43}$ | $\textbf{88.37} { \pm 1.25}$ |
> > > |Geom-GCN | $66.76 { \pm 2.72}$ | $64.51 { \pm 3.66}$ | $31.59 { \pm 1.15}$ | $38.15 { \pm 0.92}$ | $60.00 { \pm 2.81}$ | $60.54 { \pm 3.67}$ | $\textbf{78.02} { \pm 1.15}$ | ${89.95} { \pm 0.47}$ | $85.35 { \pm 1.57}$ |
> > > |PairNorm | $60.27 { \pm 4.34}$ | $48.43 { \pm 6.14}$ | $27.40 { \pm 1.24}$ | $50.44 { \pm 2.04}$ | $62.74 { \pm 2.82}$ | $58.92 { \pm 3.15}$ | $73.59 { \pm 1.47}$ | $87.53 { \pm 0.44}$ | $85.79 { \pm 1.01}$ |
> > > |GraphSAGE | $82.43 { \pm 6.14}$ | $81.18 { \pm 5.56}$ | $34.23 { \pm 0.99}$ | $41.61 { \pm 0.74}$ | $58.73 { \pm 1.68}$ | $75.95 { \pm 5.01}$ | $76.04 { \pm 1.30}$ | $88.45 { \pm 0.50}$ | $86.90 { \pm 1.04}$|
> > > |GCN | $55.14 { \pm 5.16}$ | $51.76 { \pm 3.06}$ | $27.32 { \pm 1.10}$ | $53.43 { \pm 2.01}$ | $64.82 { \pm 2.24}$ | $60.54 { \pm 5.30}$ | $76.50 { \pm 1.36}$ | $88.42 { \pm 0.50}$ | $86.98 { \pm 1.27}$ |
> > > |GAT | $52.16 { \pm 6.63}$ | $49.41 { \pm 4.09}$ | $27.44 { \pm 0.89}$ | $40.72 { \pm 1.55}$ | $60.26 { \pm 2.50}$ | $61.89 { \pm 5.05}$ | $76.55 { \pm 1.23}$ | $87.30 { \pm 1.10}$ | $86.33 { \pm 0.48}$ |
> > > |MLP | $80.81 { \pm 4.75}$ | $85.29 { \pm 3.31}$ | $36.53 { \pm 0.70}$ | $28.77 { \pm 1.56}$ | $46.21 { \pm 2.99}$ | $81.89 { \pm 6.40}$ | $74.02 { \pm 1.90}$ | $87.16 { \pm 0.37}$ | $75.69 { \pm 2.00}$ |
> > >
> > > [1] Diverse Message Passing for Attribute with Heterophily. NeurIPS 2021
> > >
> > > [2] How Powerful are Graph Neural Networks? ICLR 2019
> > >
> > > [3] Beyond Homophily in Graph Neural Networks: Current Limitations and Effective Designs. NeurIPS 2020
> > >
> > > [4] Nested Graph Neural Networks. NeurIPS 2021
> > >
> > > [5] Is Homophily a Necessity for Graph Neural Networks? ICLR 2022
> > >
> > > [6] Two Sides of the Same Coin: Heterophily and Oversmoothing in Graph Convolutional Neural Networks. ICDM 2022
> > >
> > > [7] Neural Sheaf Diffusion: A Topological Perspective on Heterophily and Oversmoothing in GNNs. NeurIPS 2022
> > >
> > > [8] Adaptive Universal Generalized PageRank Graph Neural Network. ICLR 2021

---

### Public Comment · ~Benedek_Andras_Rozemberczki1 · 2022-11-05
**Misattribution of datasets**

The paper misattributed the authorship of the Chameleons and Squirrels datasets. These datasets were proposed in this ICLR submission:

https://openreview.net/forum?id=HJxiMAVtPH

The Pei et al. paper cited by the authors took the Squirrel and Chameleons datasets and used those for benchmarking, but had nothing to do with the creation of the datasets. The correct citation for the paper which proposed the datasets is:

```bibtex
>@article{musae,
          author = {Rozemberczki, Benedek and Allen, Carl and Sarkar, Rik},
          title = {{Multi-Scale Attributed Node Embedding}},
          journal = {Journal of Complex Networks},
          volume = {9},
          number = {2},
          year = {2021},
}
```

---

> ### Author Response · Authors · 2022-11-10
> **Response to Benedek Andras Rozemberczki**
>
> Thank you for your reminder, we’ll append the citation for that paper.

---

### Author Response · Authors · 2022-11-16
**General comment**

We thank all the reviewers for their time and comments. We updated our submission and included the following main changes:

- Explain the performance of our model on the Squirrel dataset, and analyze the connection with over-squashing problem [1,2]. (in Section 3.1 and Appendix B.3)

- Report and analyze the performance of the model at 64 layers, when dealing with the over-smoothing problem. (in Appendix D)

- Added references to the baseline [3,4] that designed personalized aggregator for GNN. (in Introduction)

- Added reference to the paper [5] that help creating the Squirrel and Chameleons datasets. (in Section 3)

We hope our reply has addressed your concerns. We would appreciate to know if you have any additional questions, concerns or clarifications you would like to ask us.

[1] On the bottleneck of graph neural networks and its practical implications. ICLR 2021

[2] Understanding over-squashing and bottlenecks on graphs via curvature. ICLR 2022

[3] Graph Neural Networks with Node-wise Architecture. KDD 2022

[4] Policy-GNN: Aggregation Optimization for Graph Neural Networks. KDD 2020

[5] Multi-Scale Attributed Node Embedding. Journal of Complex Networks 2021

---

### Decision · Program_Chairs · 2023-01-20

**Decision:**

Accept: poster

**Justification For Why Not Higher Score:**

Not sufficiently novel/not sufficiently good results

**Justification For Why Not Lower Score:**

interesting idea

**Metareview: Summary, Strengths And Weaknesses:**

The paper presents an interesting variant of message passing in GNNs in which the messages are ordered. This mechanism is shown to address the problems of GNNs on heterophilic graphs. The reviewers complained about insufficient SOTA baselines, which the authors addressed in the rebuttal. We believe the paper can be accepted.

**Note From Pc:**

if the above contains the word "oral" or "spotlight" please see: "oral" presentation means -> notable-top-5% and "spotlight" means -> notable-top-25%. As stated in our emails, we are disassociating presentation type from AC recommendations